# RNA Modification Level Estimation with pulseR

**DOI:** 10.3390/genes9120619

**Published:** 2018-12-10

**Authors:** Etienne Boileau, Christoph Dieterich

**Affiliations:** 1Bioinformatics and Systems Cardiology, Klaus Tschira Institute for Integrative Computational Cardiology and Department of Internal Medicine III, University Hospital Heidelberg, 69120 Heidelberg, Germany; 2German Centre for Cardiovascular Research (DZHK), Partner Site Heidelberg/Mannheim, 69120 Heidelberg, Germany

**Keywords:** RNA-seq, confidence interval, MeRIP, m^6^A, computational biology, software

## Abstract

RNA modifications regulate the complex life of transcripts. An experimental approach called LAIC-seq was developed to characterize modification levels on a transcriptome-wide scale. In this method, the modified and unmodified molecules are separated using antibodies specific for a given RNA modification (e.g., m^6^A). In essence, the procedure of biochemical separation yields three fractions: Input, eluate, and supernatent, which are subjected to RNA-seq. In this work, we present a bioinformatics workflow, which starts from RNA-seq data to infer gene-specific modification levels by a statistical model on a transcriptome-wide scale. Our workflow centers around the pulseR package, which was originally developed for the analysis of metabolic labeling experiments. We demonstrate how to analyze data without external normalization (i.e., in the absence of spike-ins), given high efficiency of separation, and how, alternatively, scaling factors can be derived from unmodified spike-ins. Importantly, our workflow provides an estimate of uncertainty of modification levels in terms of confidence intervals for model parameters, such as gene expression and RNA modification levels. We also compare alternative model parametrizations, log-odds, or the proportion of the modified molecules and discuss the pros and cons of each representation. In summary, our workflow is a versatile approach to RNA modification level estimation, which is open to any read-count-based experimental approach.

## 1. Introduction

Until recently, mRNA modifications were underrepresented in the scientific literature. Specifically, internal modifications in mRNA and their roles in gene expression control were not considered in detail. The development of new sequencing techniques and discoveries from genetic studies have renewed the interest in understanding the effects of internal mRNA modifications [1]. Over 150 unique chemical modifications of RNA are known to date. At the time of writing, the following mRNA modifications have been studied in greater detail in eukaryotes: Inosine, m^6^A, m^5^C, m^1^A, and pseudouridine. In particular, m^6^A was the first reversible RNA modification found in eukaryotes. It can be written by several methyltransferases and erased by two known demethylases (FTO and ALKBH5) [2]. Multiple reader proteins have been identified which, together with writers and erasers, indirectly affect RNA metabolism, including maturation, translation, and decay of mRNAs [3]. The advancement of the field stalled until the arrival of transcriptome-wide mapping techniques, such as methylated RNA immunoprecipitation, followed by deep sequencing (MeRIP-seq).

In 2012, two groups independently developed MeRIP-seq methods to assess the in vivo methylation state of m^6^A sites, which allowed the genome-wide mapping of m^6^A modifications [4,5]. These methods detected over 12,000 m^6^A peaks in more than 7000 genes in human and mouse cells. These studies also showed that the methylation status of some m^6^A sites changes in response to stress. Global abundance level estimates in mRNA ranged from an m^1^A/A ratio of ≈0.02% [6,7] to an m^6^A/A ratio of ≈0.4% [8], but these number certainly depend on the cellular context.

In this manuscript, we focus on the MeRIP-seq procedure, which allows to separate modified molecules from the pool of total RNA, provided that suitable antibodies are available. A variant thereof is the recently developed m^6^A-LAIC-seq approach [9], which addresses the question of quantifying transcript m^6^A levels of particular genes or the relationship of m^6^A modification(s) to alternative RNA isoforms. To deconvolute the m^6^A epitranscriptome, mRNAs are not fragmented prior to RNA immunoprecipitation (IP), and full-length transcripts are sequenced in both m^6^A-positive and m^6^A-negative fractions post-IP.

## 2. Results

The published m^6^A-LAIC-seq dataset (NCBI accession SRP055176) consists of three fractions: The original pool, or input RNA, m^6^A modified RNA, or eluate, and m^6^A nonmodified RNA, referred to as supernatent. PolyA+ RNA was collected from two independent cell lines (GM12878 and H1-ESC) in 2 replicates and separated by antibody (Synaptic Systems, Göttingen, Germany, cat# 202 003) into the aforementioned fractions. RNA-seq libraries were prepared as described in Reference [9] and sequenced in paired-end format (2 × 50 nt) on an Illumina Hiseq 2000 (San Diego, CA, United States). To estimate modification abundance for a given RNA molecule, the different fractions (input, supernatent, and eluate) need to be normalized accordingly. It is not sufficient to measure RNA abundances in the fractions alone, as one needs to know how fractions relate to each other by assessing the efficiency of the IP. Moreover, fraction cross-contamination may affect the correctness of estimations, resulting in over- or underestimation of modification rates (specificity). One way to tackle this problem is to use synthetic RNA spike-ins. Adding them right before the IP allows to recover additional normalization factors to account for differences in the efficiency of separation (Figure 1A). One has to estimate expression and modification levels for every gene, while normalization factors are calculated from the spike-ins’ read counts.

Molinie et al. [9] used spike-in controls before anti-m^6^A IP and showed 100% efficiency in pulling down m^6^A-positive transcripts. In this case, for simplicity, we can assume that fraction cross-contamination is negligible, and we use the simpler model, where methylation level for a given gene is proportional to the mean expression (Figure 1B).

Under this scenario, if no external spike-ins are available, the mean read counts in the input RNA sample are taken as a reference and the relation between the fractions is inferred from the data, i.e., the model is identifiable as $x_{1}$ and $x_{2}$ are shared across all genes. Ideally, nonmodified spike-in sets can be used to determine the normalization constants $x_{1}$ and $x_{2}$, provided they are added in equal amounts after the IP, as in the original LAIC-seq experiment (ERCC RNA spike-In mix was added to each fraction in equal amount right after the purification, allowing to recover the relation between the modified and nonmodified RNA fractions). We thus consider two scenarios, without and with external spike-ins, where normalization constants are either inferred from the data or estimated using the spike-ins’ read counts. In the first case, sequencing of the input RNA fraction is required, while only the sequencing of eluate and supernatent fractions is necessary for the latter.

In the following, we guide the reader through a bioinformatics workflow to estimate m^6^A levels from a standard LAIC-seq experiment, assuming high efficiency of separation (Figure 1B), using the pulseR package [10].

### 2.1. Read Processing and Mapping

All sequencing data were obtained from the original distribution through the short read archive (accession SRP055176) at the National Center for Biotechnology Information (NCBI). We removed adapter sequences and quality clipped reads using FLEXBAR 3.0.3 [11]. Ribosomal contaminant sequences (nuclear encoded rRNA) were subtracted from the pool of reads by mapping against the 45S rRNA gene cluster with Bowtie2 v2.3.0 [12]. All remaining reads were aligned against the human reference genome (EnsEMBL 90) using STAR 2.5.3 [13]. Gene and transcript count tables were estimated with Stringtie 1.3.3b and the complementary prepDE script [14]. For all analyses, we required mean read count across input samples > 100 or, when using spike-ins, mean read count across all eluate and supernatent samples > 100.

### 2.2. m^6^A Estimates Using the Ratio of RNA Abundances

We refer to this approach as the LAIC-seq method. m^6^A levels for each gene were quantified as in Molinie et al. using the fragment counts of the ERCC RNAs [9]. A log–log linear regression was fitted to estimate the intercept R for each of the four corresponding sample sets (two per cell type), such that m^6^A levels were calculated as follows:(1)$$m^{6}A=\frac{E}{E+S\cdot 2^{R}},$$ using the eluate (*E*) and supernatent (*S*) counts (Appendix A). The model was fitted only using ERCC RNAs with at least 100 counts. We empahize that this approach does not take into account any particular aspects of the underlying sequencing data. With the pulseR package, by contrast, abundance levels are estimated assuming a negative–binomial distribution in maximum likelihood of all parameters, and accounting for differences in sequencing depth between samples, further allowing to obtain modification levels in the absence of spike-ins.

### 2.3. m^6^A Estimates Using pulseR

The pulseR package was developed specifically for metabolic labeling experiments, but can easily be adapted to epitranscriptomics workflow, where RNA modification levels are estimated from read count data. Intuitively, the RNA methylation level is represented as a proportion α of all molecules in the input sample. In this case, α∈[0,1]. Alternatively, this can be represented as the logit of the methylation level, which is the logarithm of the odds, or $\gamma =\mathrm{logit}\left(\alpha \right)=\log\frac{\alpha }{1-\alpha }$, which ranges from (-∞,+∞).

In pulseR, samples are normalized for sequencing depth, and normalization between fractions is performed during the fitting procedure or else directly estimated from spike-in read counts. To simplify the notation, in the following sections, we do not include the normalization constants $x_{1}$ and $x_{2}$ in the equations. We used the parametrization based on the logit transformation. For the fitting procedure, we kept gene-specific parameters at the same (logarithmic) scale. An introductory workflow using the pulseR package with a subset of the supplementary data from the original LAIC-seq experiment is available at https://dieterich-lab.github.io/pulseR/articles/epitranscriptomics.html. We also provide the complete workflow in the supplementary files, including data and analysis scripts, as well as a full comparison between pulseR and the published estimates from Molinie et al. [9].

In the next section, we consider the case of m^6^A level analysis in the absence of spike-ins.

#### 2.3.1. Estimates Without Spike-Ins

For the analysis, we write the model in Figure 1B into a set of logit-transformed formulas, representing read counts for RNA input, modified and nonmodified RNA.
(2)$$\left[input\right]=e^{\mu }$$
(3)$$\left[eluate\right]=e^{\mu +\gamma }/\left(1+e^{\gamma }\right)$$
(4)$$\left[\mathrm{supernatent}\right]=e^{\mu }/\left(1+e^{\gamma }\right)$$

Additional information regarding counts, sample groupings, and how to specify boundaries on parameters and usage can be found in the pulseR documentation (https://dieterich-lab.github.io/pulseR/). In summary, the model is fitted in several steps as follows:Fitting of gene specific parameters: Expression level μ and logit(α), referred to as γ (or log-odds of methylation);fitting of normalisation factors (for a spike-ins free design): These are shared between the samples from the same fraction;fitting of global parameters: Size parameter for the negative binomial distribution (overdispersion in read counts).

To make model fitting most efficient, it is helpful to initialize the parameters close to their expected values. For example, the initial parameter value for μ should be set close to the logarithm of genes read counts in the total fraction.

We performed this procedure for the whole data set, ignoring the ERCC spike-in information. Results were compared to those obtained with the LAIC-seq method (Figure 2), presented in terms of methylation level and their logit transformation. In the absence of spike-ins, pulseR was able to recover the fraction scaling factors such that the model fit yielded values that were significantly close to those obtained with Equation (Equation 1), determined using ERCC counts only. However, we observed a global bias between the two methods, which was particularly visible for replicate 2 of both cell lines (Figure 3 and Appendix A).

#### 2.3.2. Confidence Intervals

An additional feature of our workflow, which is not readily available with the LAIC-seq method, is the computation of confidence intervals for the estimated model parameters (Figure 3). Confidence intervals support the user in assessing the uncertainty and quality of point estimates. For example, point estimates of RNA modification levels could very well differ across cell types (or experimental conditions in general), but their respective confidence intervals (or probable range of values) overlap. In that case, the observed difference is not statistically significant. This is especially true for lowly covered and highly variable data sets that lead to point estimators with wide confidence intervals.

In the pulseR package, the function ciGene allows to estimate confidence intervals for the gene-specific parameters, such as expression and modification levels. In this example, the uncertainty of normalization factors was neglected and the profile of the likelihood (see https://dieterich-lab.github.io/pulseR/articles/confidence.html) was computed, assuming that the normalization factors and size parameter were fixed.

#### 2.3.3. Estimates with Spike-Ins

If external spike-ins are available, we can use this information to determine the normalization factors, speeding up the fitting procedure. In summary, the input RNA is neglected, and spike-ins are used to directly relate eluate and supernatent fractions. This is possible because equal amounts of ERCC RNA spike-In mix were added to both fractions after IP. These counts are thus added to determine the scaling factors (without fitting). The formulas then reduce to:(5)$$[eluate]=e^{\mu +\gamma }/\left(1+e^{\gamma }\right),$$
(6)$$\left[\mathrm{supernatent}\right]=e^{\mu }/\left(1+e^{\gamma }\right).$$

To use this information in pulseR, it suffices to pass the list of spike-ins as an additional argument, after adding the spike-ins to the count table with other genes. One of the fractions (e.g., supernatent) must be assigned as a reference. As seen in Figure 4, pulseR estimates correlated very well with the LAIC-seq method. When considering each sample individually, the fit was perfect and we observed no global bias (Appendix A). Similar to the case without spike-ins, confidence intervals for γ were also calculated (available in the Appendix A).

## 3. Discussion

We have presented a comprehensive workflow to assess the modification status of RNA molecules on a transcriptome-wide scale. We have demonstrated our approach based on a published LAIC-seq experiment, which involves an antibody-driven biochemical separation step on intact (i.e., not fragmented) RNA. Each input sample was separated into an eluate (modified) and supernatent (nonmodified) fraction. Target molecules of interest were turned into a table of read counts by an RNA sequencing experiment, which we had reanalyzed from raw data. Contrary to previous approaches, such as in Molinie et al. [9], we set up a statistical model with the R package pulseR [10], which respects the underlying nature of the data in a principled way [15].

We used a simple model to estimate the RNA methylation level, using a logit-transform in the absence of cross-contamination in two alternative scenarios: (a) Without spike-ins and (b) with ERCC spike-ins, to normalize read counts across fractions. In both scenarios, we showed that our estimates correlate very well with those obtained with the LAIC-seq method, and also with the data published in Molinie et al. [9], when considering the common set of genes for which methylation levels were available. When no spike-ins were used, a global bias was observed in some cases, due to the estimation of the scaling factors, including normalization constants for sequencing depth in pulseR. The LAIC-seq method yielded higher methylation levels when compared to those obtained with pulseR. A unique advantage of our workflow is the computation of confidence intervals for model parameters, such as RNA modification levels.

Our workflow is applicable to any current and future technique based on read counts from both fragmented and nonfragmented RNA, which can be separated into modified and unmodified fractions by biochemical means, chemical, or metabolic RNA labeling.

References

## Figures and Tables

**Figure 1 genes-09-00619-f001:**
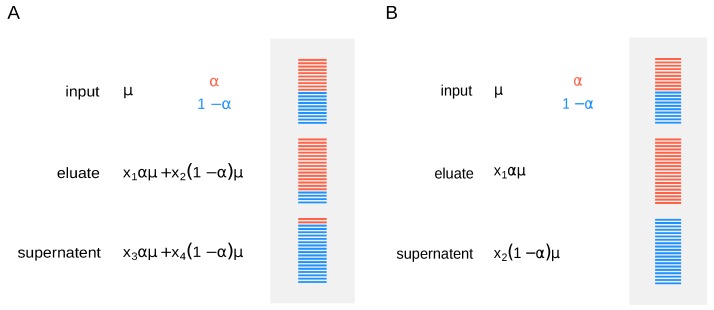
Experimental design and modeling of a quantitative MeRIP experiment. Methylated RNA species are shown in red and unmethylated in blue. μ is the RNA abundance, α is the proportion of methylated RNA, and 1-α of unmethylated RNA. Normalization constants are denoted $x_{1}$ to $x_{4}$.

**Figure 2 genes-09-00619-f002:**
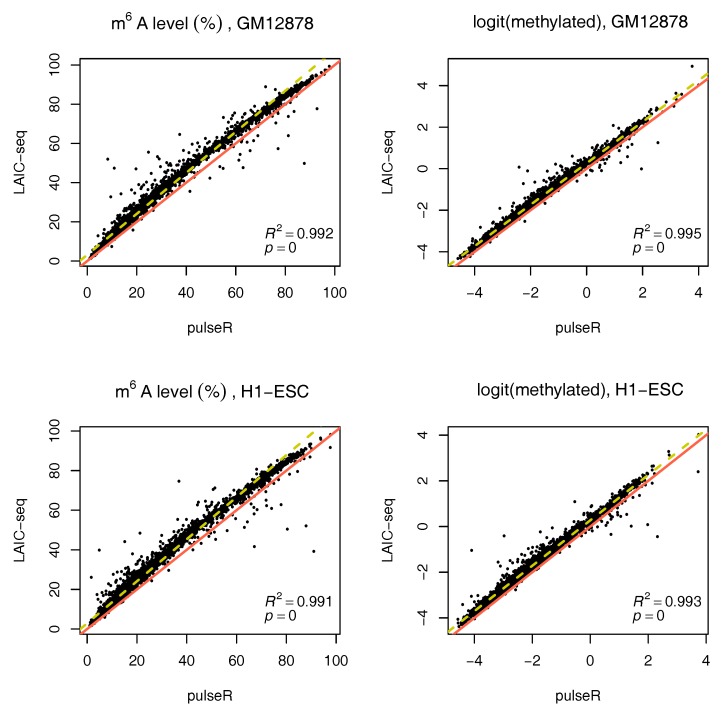
Comparison of m^6^A methylation levels between pulseR estimates and the LAIC-seq method for both cell lines without spike-ins, represented as a percentage (**left**) or using log-odds (**right**). Line of equality in red, linear fit in yellow with coefficient of determination.

**Figure 3 genes-09-00619-f003:**
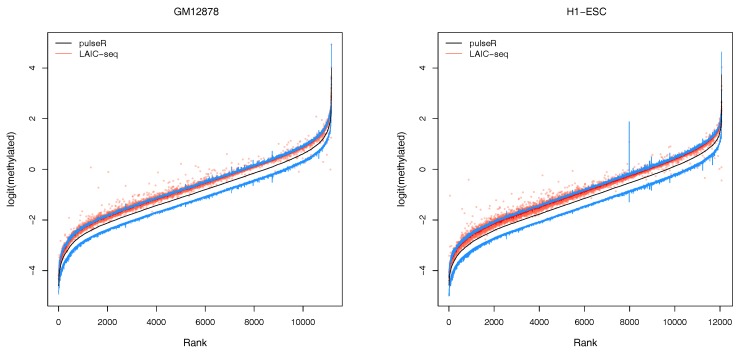
Confidence intervals for GM12878 and H1-ESC cell lines, without spike-ins. Genes are sorted by their log-odds of methylation γ (black line). Estimates obtained with the LAIC-seq method are shown as red dots with a running median. Upper and lower 95% confidence interval boundaries are show in blue (above/below mean value in black).

**Figure 4 genes-09-00619-f004:**
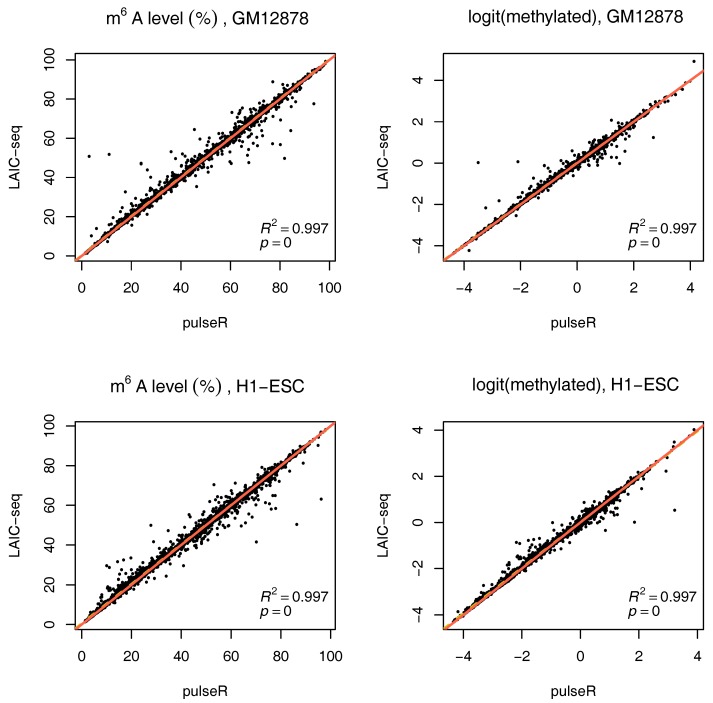
Comparison of m^6^A methylation levels between pulseR estimates and the LAIC-seq method for both cell lines using ERCC spike-in data, represented as a percentage (**left**) or using log-odds (**right**). Line of equality in red, linear fit in yellow with coefficient of determination.

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
