# Peer review of "RNA Modification Level Estimation with pulseR"

_genes, 2018, doi:10.3390/genes9120619_

Round 1
Reviewer 1 Report
In this manuscript, the authors a workflow to analyzed MeRIP-seq data. Although the authors provide the detailed information about the pipelines and algorithms implemented, the authors only compared to the published results of the MeRIP-seq data were generated. In other words, it is not immediately clear the advantage of using the authors’ workflow over the published data analysis pipeline. Thus, it is mandatory to confirm their analyzed results by actual biological experiments by knocking down the m6A enzymes and perform an alternative way of confirming the m6A sites. Or take such MeRIP-seq data to further analyze the data using their methods. More specific comments are listed below:
Major points:
[1] Lines 79-80. Given that many rRNAs are methylated, why do the authors remove such sequences?
[2] The authors provide their codes at this URL: https://dieterich-lab.github.io/pulseR/articles/epitranscriptomics.html. However, the actual author for these codes, Uvarovskii Alexey, is not listed as co-author on this manuscript, which raises a significant scientific concern.
Minor points:
(1) It is unusual to have references in the Abstract section.
(2) Line 26: “ RNA-binding proteins place”. What do authors mean by “place”?
(3) Lines 26-28. This is an overstatement
(4) Having a figure in the Introduction section is unusual.
(5) Line 52: How is it possible to judged for the efficiency of IP?
Author Response
Manuscript ID: genes-395734
Type of manuscript: Article
Title: RNA modification level estimation with pulseR
Authors: Etienne Boileau, Christoph Dieterich
Received: 9 November 2018
3 December 2018
Subject: Submission of revised manuscript genes-395734
We would like to thank the reviewers for their suggestions. We have carefully reviewed the comments and have revised the manuscript accordingly. Our response is detailed below. In particular, in view of one comment made by reviewer 2, we have now included a complete analysis based on our read processing results only, and made the comparison with the published estimates available as supplement. Additional figures were also added as supplementary material, as well as all code/scripts and data. Accordingly, some major changes were made to the manuscript. We have highlighted in red those which responded to specific comments.
We hope the revised version is now suitable for publication.
Response to Reviewer 1:
[...] it is not immediately clear the advantage of using the authors’ workflow over the published data analysis pipeline. Thus, it is mandatory to confirm their analyzed results by actual biological experiments by knocking down the m6A enzymes and perform an alternative way of confirming the m6A sites.
Compared to the published analysis pipeline, we propose a relatively straightforward workflow, with the advantage (1) that we are able to quantify RNA methylation level using different parametrization without requiring additional spike-ins, provided high efficiency of separation; and (2) that we provide confidence intervals with our estimates, which is not the case for any published workflow. Admittedly, if no spike-ins are used, we require the sequencing of total RNA, which is not the case if such spike-ins are available. In both cases, though, we showed that our estimates correlate very well with those obtained with the published method, as well as with the published estimates. We would also like to emphasize that m6A-levels were estimated in [9] based on a log-log linear fit for each of the four corresponding sample sets (two per cell type). This approach does not consider the underlying nature of sequencing data, which we do by estimating abundance levels assuming the negative-binomial distribution in maximum-likelihood of all parameters, and by accounting for differences in sequencing depth between samples. Our workflow also allows to work directly with count data. Thus, generally, we think that our workflow may provide some advantages over existing pipelines, and given that our results are consistent with those published in [9] (which in turn were validated with methylated spike-ins), we do not think that confirming these with additional experiments would add any substance to the manuscript.
Lines 79-80. Given that many rRNAs are methylated, why do the authors remove such sequences?
The reviewer is right that RNA modifications also occur in rRNAs, as well as in tRNAs and others. In this study, however, we are particularly interested in comparing mRNA methylation levels with those published in [9], the reason being that the data consist of polyadenylated RNAs. There is thus no justification to include ribosomal sequences (in fact not many reads mapped to ribosomal sequences, see multiqc_report.html). In this context, we remove unwanted RNA species, as indicated in our read mapping protocol.
[…] the actual author for these codes, Uvarovskii Alexey, is not listed as co-author on this manuscript, which raises a significant scientific concern.
This has been addressed with the editors, and we believe that all contributions have now been clarified.
It is unusual to have references in the Abstract section.
References in the abstract were removed.
Line 26: “ RNA-binding proteins place”. What do authors mean by “place”?
This sentence has been clarified in the text.
Lines 26-28. This is an overstatement
See previous comment.
Having a figure in the Introduction section is unusual.
This part of the manuscript was restructured accordingly.
Line 52: How is it possible to judged for the efficiency of IP?
We refer the reviewer to [9], and in particular to Methods, Figure 1 and Supplementary Figure 1. In particular, the authors used spike-in controls after poly(A)-RNA selection, but before anti-m6A IP, with both modified and nonmodified transcripts. They showed that their protocol yielded 100% efficiency in pulling down m6A-positive transcripts.
Reviewer 2 Report
The authors recently developed a tool, pulseR (Uvarovskii A and Dieterich C, Bioinformatics 2017), for the analysis of RNA metabolic labeling data. pulseR can be used for the integrative analysis of nascent and total RNA-seq data, to provide the kinetic rates of RNA synthesis and degradation. To this end, before modeling the kinetic rates, the tool has to deal with the normalization of nascent and total datasets, and gene-specific scaling factors have to be determined for each sample. The authors recognized that this issue is analogous to the task of quantifying the proportion of m6A-methylated transcripts following the IP with an m6A-sepecific antibody, where the total vs nascent RNAs comparison is now total vs methylated RNAs.
pulseR has thus been applied for the analysis of m6A-LAIC-seq (from here on “LAIC”) data, which provides gene-specific % of methylation based on the sequencing of non-fragmented methylated and un-methylated RNAs. The estimates nicely fit with the original LAIC estimates (which in turn were nicely validated with a pool of methylated spike-ins). The advantage of using pulseR is two fold: (1) it allows quantifying % of RNA methylation without requiring the addition of spike-ins (while it does so by requiring the sequencing of total RNA, which is not necessary in LAIC), and provides confidence intervals of those estimates, which are particularly precious for differential analyses; (2) if spikein were used, it quantifies the % of RNA methylation including confidence intervals. Importantly, confidence intervals could not be obtained using the original LAIC analysis methodology.
Major revision:
I would urge the authors to release the actual R code used to produce the manuscript figures, maybe as supplementary file. This would also increase the impact of this study.
Few additional clarifications are needed (minor revisions):
Eq.(1): the authors should clarify that the sequencing of total RNA is required if using pulseR without spikeins (LAIC does require spikeins, but it does not require the sequencing of total, while total RNA was sequenced in that study).
Why the “x” normalization factors in Fig 1B are not in the equations (1-3), while they are discussed in the modeling steps (lines 101-102)?
Are those steps (lines 99-104) executed in parallel or in a particular order?
The (small) shift, i.e. underestimation, compared LAIC published estimates, should be briefly discussed. Is there any reason that this is consistently an underestimation in the two cell lines?
A threshold of 100 read counts used as a filter of expression seems particularly conservative. How many genes are excluded with this threshold and how the correlation with LAIC data would be with lower thresholds?
Why excluding genes that were reported having LAIC estimates equal to 0 (unmethylated transcripts)? This is an important piece of data in the field of m6A.
Figure 3 is not cited in the text.
Line 130: “input RNA is neglected, since we cannot relate it to the other fractions” [when using spikeins]. I think it should be “it is not necessary” rather that “we cannot relate it”. If total can be related to the other samples without spikeins, I do not see why this should not be possible with the addition of spikeins. Rather, it is not need, as spikeins can be used to directly relate methylated and unmethylated fractions.
When using spikeinds, could the sequencing of the unmethylated fraction be replaced by the sequencing of the total RNA sample (since unmethylated could be recovered as total – methylated via spikeins normalization)?
Are confidence intervals provided in the spikein-based approach?
Line 150: I think MeRIP should be replace by m6A-LAIC, since on line 162 it is stated that this applies only to the analysis of non-fragmented RNA.
Line 162: could the authors briefly discuss why this method is not suitable for the analysis of fragmented data, such as MeRIP-seq data?
Author Response
Manuscript ID: genes-395734
Type of manuscript: Article
Title: RNA modification level estimation with pulseR
Authors: Etienne Boileau, Christoph Dieterich
Received: 9 November 2018
3 December 2018
Subject: Submission of revised manuscript genes-395734
We would like to thank the reviewers for their suggestions. We have carefully reviewed the comments and have revised the manuscript accordingly. Our response is detailed below. In particular, in view of one comment made by reviewer 2, we have now included a complete analysis based on our read processing results only, and made the comparison with the published estimates available as supplement. Additional figures were also added as supplementary material, as well as all code/scripts and data. Accordingly, some major changes were made to the manuscript. We have highlighted in red those which responded to specific comments.
We hope the revised version is now suitable for publication.
Response to Reviewer 2:
I would urge the authors to release the actual R code used to produce the manuscript figures, maybe as supplementary file. This would also increase the impact of this study.
The pulseR package is already under a GPL-3 license here https://github.com/dieterich-lab/pulseR. An example workflow applied to epitranscriptomics is also available here https://dieterich-lab.github.io/pulseR/articles/epitranscriptomics.html. This information is already available in the manuscript. We will also release as supplementary material the scripts used to generate the data and the figures as well as the mapping data.
Eq.(1): the authors should clarify that the sequencing of total RNA is required if using pulseR without spikeins (LAIC does require spikeins, but it does not require the sequencing of total, while total RNA was sequenced in that study).
This has been clarified in the text.
Why the “x” normalization factors in Fig 1B are not in the equations (1-3), while they are discussed in the modeling steps (lines 101-102)?
This has been clarified in the text.
Are those steps (lines 99-104) executed in parallel or in a particular order?
These are performed iteratively until convergence criteria are met, as specified by the user. The reviewer is referred to [10].
The (small) shift, i.e. underestimation, compared LAIC published estimates, should be briefly discussed. Is there any reason that this is consistently an underestimation in the two cell lines?
We believe this has now been addressed in the manuscript. We have re-written the analyses based on a comparison between pulseR and the “LAIC-seq method”, where we have estimated m6A levels using the published methodology with our counts. This avoids the pitfalls of comparing datasets obtained with different read processing pipelines. The comparison with the published estimates is still available as supplementary material.
In summary, there is a global bias, particularly for replicate 2 (supplementary figures), largely due to the estimation of the normalisation factors, when ignoring spike-ins in the pulseR workflow.
A threshold of 100 read counts used as a filter of expression seems particularly conservative. How many genes are excluded with this threshold and how the correlation with LAIC data would be with lower thresholds?
Although it is admittedly conservative, we do not think this affects the validity of our results. It is common in differential gene expression analysis to filter low-expression genes to increase sensitivity and precision.
In [9], they used a threshold of 50 in both replicates for either eluate or supernatant counts, whereas to fit the regression model, a threshold of 100 (ERCC counts) was used. We also used a threshold of 100 counts to fit the regression model for the “LAIC-seq” methodology.
For the fitting procedure, we used a threshold of 100 counts, however this was done using the mean counts per gene (and ERCC if using spike-ins) across all relevant samples. Similarly, in the first scenario (without spike-ins), we used the mean count per gene across the total/input samples. This has now been clarified in the text.
For GM12878, filtering at a threshold of 10, 50 and 100, as explained above, resulted in filtering 47.5%, 58.5% and 62.9% of lowly abundant genes. For H1-ESC, this resulted in filtering 44.6%, 56.1% and 61.1% of lowly abundant genes. Comparatively, filtering the published estimates as explained above resulted in filtering 52.1% and 48.7% of lowly abundant genes, for GM12878 and H1-ESC, respectively. Overall, this is not drastically different.
We nevertheless compared pulseR estimates with the published estimates using a threshold of 10 and 50, as explained above. The correlation, as expected, is lower using a threshold of 10, and a global bias is also visible:
GM12878 m6A level R2=0.907 | GM12878 logit(methylated) R2=0.933 | |
H1-ESC m6A level R2=0.917 | H1-ESC logit(methylated) R2=0.922 |
But for a threshold of 50:
GM12878 m6A level R2=0.93 | GM12878 logit(methylated) R2=0.939 | |
H1-ESC m6A level R2=0.935 | H1-ESC logit(methylated) R2=0.937 |
However, as explained, we now compare pulseR estimates and those obtained with the “LAIC-seq method” on the same read count dataset.
Why excluding genes that were reported having LAIC estimates equal to 0 (unmethylated transcripts)? This is an important piece of data in the field of m6A.
This comparison is not part of the manuscript anymore, but available as supplement. We did remove missing entries from the published data in [9], and filtered the data using a threshold of 50 in both replicates for either eluate or supernatent counts, but did not explicitly filtered out those with zero methylation level.
Figure 3 is not cited in the text.
A cross-reference has been added.
Line 130: “input RNA is neglected, since we cannot relate it to the other fractions” […] Rather, it is not need, as spikeins can be used to directly relate methylated and unmethylated fractions.
This has been clarified in the text, however see comment below.
When using spikeins, could the sequencing of the unmethylated fraction be replaced by the sequencing of the total RNA sample (since unmethylated could be recovered as total – methylated via spikeins normalization)?
In theory, yes, if the input fraction originates from the same sample as eluate and supernatant. However, we have tried to replace the sequencing of the unmethylated fraction with the input fraction and obtained worse results.
Since we are working with published data, we would like to present the data in the original context. The reviewer is more than welcome to comment on this matter.
Are confidence intervals provided in the spikein-based approach?
Yes. Supplementary figures/notebooks include the confidence intervals in the spike-in based approach.
Line 150: I think MeRIP should be replace by m6A-LAIC, since on line 162 it is stated that this applies only to the analysis of non-fragmented RNA.
This has been modified, however see comment below.
Line 162: could the authors briefly discuss why this method is not suitable for the analysis of fragmented data, such as MeRIP-seq data?
We focused our comparison on the m6A-LAIC setting where RNA isoform level information is preserved. However, our workflow is not limited to this application, and we can as well apply it to the more general case of MeRIP-seq data.
Round 2
Reviewer 1 Report
I have no further comment to make.